# Fiber Loop Mirror Based on Optical Fiber Circulator for Sensing Applications

**DOI:** 10.3390/s23020618

**Published:** 2023-01-05

**Authors:** Paulo Robalinho, Beatriz Soares, António Lobo, Susana Silva, Orlando Frazão

**Affiliations:** 1Institute for Systems and Computer Engineering, Technology and Science (INESC TEC), 4169-007 Porto, Portugal; 2Faculty of Engineering, Porto University, 4200-465 Porto, Portugal; 3Faculty of Health Sciences, University Fernando Pessoa, 4200-150 Porto, Portugal

**Keywords:** optical fiber sensor, fiber loop mirror, birefringence, strain sensor, torsion sensor, optical circulator

## Abstract

In this paper, a different Fiber Loop Mirror (FLM) configuration with two circulators is presented. This configuration is demonstrated and characterized for sensing applications. This new design concept was used for strain and torsion discrimination. For strain measurement, the interference fringe displacement has a sensitivity of (0.576 ± 0.009) pm‧με^−1^. When the FFT (Fast Fourier Transformer) is calculated and the frequency shift and signal amplitude are monitored, the sensitivities are (−2.1 ± 0.3) × 10^−4^ nm^−1^ με^−1^ and (4.9 ± 0.3) × 10^−7^ με^−1^, respectively. For the characterization in torsion, an FFT peaks variation of (−2.177 ± 0.002) × 10^−12^ nm^−1^/° and an amplitude variation of (1.02 ± 0.06) × 10^−3^/° are achieved. This configuration allows the use of a wide range of fiber lengths and with different refractive indices for controlling the free spectral range (FSR) and achieving refractive index differences, i.e., birefringence, higher than 10^−2^, which is essential for the development of high sensitivity physical parameter sensors, such as operating on the Vernier effect. Furthermore, this FLM configuration allows the system to be balanced, which is not possible with traditional FLMs.

## 1. Introduction

The Fiber Loop Mirror (FLM) is one of the most flexible configurations in optical systems. With applications in lasers, as well as in sensors, they can be used as mirrors [1] or as optical filters [2]. As a mirror, usually implemented in mode-locked fiber lasers [3], two rings are formed by connecting the coupler’s parallel ports [4] or cross ports [5], where at least one of them is composed of a nonlinear medium. These configurations are called Nonlinear Optical Loop Mirrors (NOLM). As a mirror device, this can also be constituted of just a ring without a nonlinear medium, where the reflectivity is provided by polarization control [6].

As an optical filter, the FLM is only composed of a ring containing a piece of high-birefringence (Hi-Bi) optical fiber [7]. Through a polarization controller, the clockwise (CW) and counterclockwise (CCW) propagation modes acquire different phases due to the birefringence of the fiber, allowing the generation of an interference pattern when they are overlapped in the coupler. When the Hi-Bi optical fiber is exposed to a temperature or strain variation, it implies a variation of the phase difference between the two beams and a spectrum variation is consequently obtained. This allows the application of FLMs as intensity or interferometric sensors for either single or simultaneous measurements, exhibiting a maximum strain sensitivity of the order of tens of pm/με and a temperature sensitivity of a few nm/°C. In addition, they can also be used as optical gyroscopes [8].

With the emergence of the Vernier effect applied to optical sensors in 2011 [9], several optical fiber sensors were used, and the FLM was no exception [10]. This phenomenon consists of the overlapping of the interference fringes of two interferometers, whose optical path difference is close to each other or close to multiple originating two waves: the carrier and the envelope [11]. Usually, sensors work based on the traditional or enhanced Vernier effect [12], where the envelope variation is measured, as it is the one that shows the highest sensitivity. In 2020, with an FLM of two Hi-Bi fibers, a new record was achieved, a sensitivity of 10 000 pm/με, where the “push-pull” concept was implemented in order to achieve the enhanced Vernier effect in an optimized way [13]. This required the implementation of two Hi-Bi fibers with a length longer than 1 m so that it was possible to achieve the colossal sensitivity with a 100 nm bandwidth erbium source. Despite the colossal sensitivities achieved, the need for interferometers with low Free Spectral Range (FSR) implies the implementation of long fibers due to the low birefringence of the current Hi-Bi fibers, where solid fibers have a birefringence of the order of 10^−4^ and PCF fibers have a birefringence of the order of 10^−3^.

This work consists of exploring a new FLM configuration with two circulators in series. This new configuration allows a typical ring interferometer to be balanced, which is not the case with traditional FLMs. The difference in arms is 20 mm in distinct output ports. In addition, this system is more versatile, since the FSR of the system also depends on the length difference of the two single-mode fibers, a situation that does not occur in traditional FLMs. Furthermore, FSR also depends on the difference in the refractive indices of the two optical fibers. The torsion and strain application will result in the change of distinct spectrum features, allowing the simultaneous measurement of the two physical quantities.

## 2. Theoretical Consideration

In the presented system, the FLM has two circulators and two single-mode optical fibers (Figure 1). The beam is first divided in two at the fiber coupler, CW and CCW. Because of the isolation of the optical circulators, the CCW beam only travels along the bottom optical fiber (forming the inner ring) and the CW beam travels along the top fiber (forming the outer ring). In addition, a polarization controller is also used to ensure maximum interference [14,15].

The electric field of the system’s arms can be described as:(1)$$E_{\mathrm{CCW}}\propto E_{\mathrm{in}}\left(E^{i\beta_{\mathrm{CCW}}\;L_{\mathrm{CCW}}}\right)/\sqrt{2}$$

And
(2)$$E_{\mathrm{CW}}\propto E_{\mathrm{in}}\left(E^{i\beta_{\mathrm{CW}}\;L_{\mathrm{CW}}}\right)/\sqrt{2}$$
where *E*_in_ is the initial electric field amplitude, *β* is propagation constant and *β* = 2π*n*/*λ*, *n* is the refractive index, *λ* is wavelength and *L* is the fiber length.

The intensity is given by:(3)$$I_{\mathrm{norm}}\propto \left(E_{\mathrm{CCW}}+E_{\mathrm{CW}}\right){\left(E_{\mathrm{CCW}}+E_{\mathrm{CW}}\right)}^{*}$$

As seen in Figure 2, the interference of two waves results in a spectrum that can be described as a cosine. For this case, the spectral intensity is given by [16]:(4)$$I_{\mathrm{norm}}\propto \cos^{2}\left(\frac{\pi \left(L_{\mathrm{CCW}}n_{\mathrm{CCW}}-L_{\mathrm{CW}}n_{\mathrm{CW}}\right)}{\lambda }\right)$$

Thus, the system has a spectrum similar to an optical cavity. This system allows measuring the difference in torsion and strain between the arms where the opposite arm to the action of the physical quantity is the reference. Thus, in the CCW optical path, the strain variation is applied and is the reference for the torsion measurement. The torsion variation in the CW optical path is applied and is the reference for strain measurements. The variation in torsion will imply a variation in interferometer visibility due to the polarization rotation of one light beam. The variation in strain will change the optical path difference, which will result in a frequency variation associated with the interference pattern, typically studied by varying the spectrum extremes.

The general equation to discriminate two distinct parameters, the frequency shift of the FFT maxima (Δ*f*), and the corresponding amplitude variation (Δ*A*), is
(5)$$\left[\begin{array}{c} \Delta f\\ \Delta A \end{array}\right]=\left[\begin{array}{cc} K_{\varepsilon ,f} & K_{\tau ,f}\\ K_{\varepsilon ,A} & K_{\tau ,A} \end{array}\right]\left[\begin{array}{c} \Delta \varepsilon \\ \Delta \tau \end{array}\right]$$
where *K_i,j_* with *i* = *ε,τ* and *j* = *f,A* are the linear fit slope, that, for this system, are shown on Section 3.2. So, the system equation is
(6)$$\left[\begin{array}{c} \Delta \varepsilon \\ \Delta \tau \end{array}\right]=\frac{1}{K_{\varepsilon ,f}K_{\tau ,A}-K_{\tau ,f}K_{\varepsilon ,A}}\left[\begin{array}{cc} K_{\tau ,A} & -K_{\tau ,f}\\ -K_{\varepsilon ,A} & K_{\varepsilon ,f} \end{array}\right]\left[\begin{array}{c} \Delta f\\ \Delta A \end{array}\right]$$

## 3. Results

The system was characterized for strain and torsion variations. This section will consist of three parts: test overview, characterization with a Δ*L* = 20 mm and Δ*n* = 0, and with Δ*L* ≈ 0 and Δ*n* ≠ 0, where Δ*L* is the length difference between two arms and Δ*n* is refraction indices difference between the two arms of the interferometer.

### 3.1. Test Overview

The experimental setup, based on Figure 1, has the ability to measure strain and tor-sion. This system was composed of a 3 dB optical coupler (2 × 2), two optical circulators, a polarization controller, standard optical fiber (SMF28e) and PS1250/1550 optical fiber, an erbium source with a 70 nm bandwidth and centered at 1565 nm, and an optical spectrum analyzer (OSA) (model YOKOGAWA AQ5370C) with a resolution of 0.02 nm; connections were made with a conventional splice machine (model Sumitomo Electric Type-72C, Osaka, Japan).

All measurements were made at a room temperature of 20 °C. For strain characterization, a conventional micrometric translation stage was used. The stage measurement step is 50 µm and the fiber length where the strain was applied was 430 ± 1 mm, so the strain measurement step was 116 µε. The strain measurement range was between 0 mε and 1.67 mε. For torsion characterization, a torsion stage was used. The measuring step was 5°, the torsion measure was between 0° and 180° and the fiber length where the torsion is applied was 50 ± 1 mm.

### 3.2. For ΔL = 20 mm and Δn = 0

In the first step, the system was set up in such a way that the refractive index of the interferometer arms was equal and the difference in arm lengths between the two circulators was 20.2 ± 0.1 mm. Each physical quantity was measured based on the perturbation of only one of the system’s arms, where strain was applied to the CCW state and torsion was applied to the CW state. Figure 3 shows the output spectrum of the system and its FFT.

In the characterization, both the frequency shift of the FFT maxima (Δ*f*) and the variation of the corresponding intensities amplitude (Δ*A*) were determined. In addition, the interference fringe shift was also studied when the strain was varied. The sensitivities associated with the data presented in Figure 4 are shown in Table 1. All of the linear fits have an R-squared value (*r*^2^) equal to or higher than 0.99.

In the strain characterization, a spectral frequency variation of (−2.1 ± 0.3) × 10^−4^ nm^−1^ με^−1^ (Figure 4a) and an amplitude variation of (4.9 ± 0.3) × 10^−7^ με^−1^ (Figure 4b) were achieved. In the torsion characterization, a spectral frequency variation of (−2.177 ± 0.002) × 10^−12^ nm^−1^/° (Figure 4c) and an amplitude variation of (−1.02 ± 0.06) × 10^−3^/° for a range of 0° to 45° were calculated (Figure 4d). In addition, a cosine amplitude variation for a range of 0° to 180° (Figure 4d) was obtained, an expected variation with the linear polarization rotation. In both cases, we can see low cross-sensitivity between signal amplitude (strain) and frequency (torsion). The pattern fringe in wavelength shift for strain and torsion was measured, and sensitivities of (0.576 ± 0.009) pm‧με^−1^ and (2.09 ± 0.06) pm/° were determined (Figure 5).

With these values results Equation (7), the matrix of this system that allows the simultaneous measurement of strain and torsion.
(7)$$\left[\begin{array}{c} \Delta \varepsilon \\ \Delta \tau \end{array}\right]=\frac{1}{2.142\times 10^{-7}}\left[\begin{array}{cc} -1.02\times 10^{-3} & 2.177\times 10^{-12}\\ -4.9\times 10^{-7} & -2.1\times 10^{-4} \end{array}\right]\left[\begin{array}{c} \Delta f\\ \Delta A \end{array}\right]$$

To demonstrate the equation, several simultaneous measurements were obtained and the equation was applied (see Figure 6). In this system, the frequency variation of the spectral fringes occurs only with the variation of the strain. The amplitude variation depends on both the torsion and the strain, being strongly dependent on the torsion when compared to its dependence on the strain. Thus, with the application of Equation (7), it is possible to achieve simultaneous strain and torsion measurements.

A phase variation that can be characterized through the common analysis of the shift of the spectra’s extremes or based on the variation of the FFT peak frequencies. From Table 1, it can be noted that the frequency shift achieved with strain variation is 100 dB higher when compared to the torsional variation of the system. The amplitude variation of the FFT peaks, which correspond to the visibility of the spectrum, achieved in the torsional variation, is 33 dB higher when compared to the strain variation of the system. While strain measurement is an interferometric measurement, since there is a change in the fringe’s spectral frequency (i.e., variation of the extremes of the spectrum), torsion measurement is an intensity measurement, since there is a linear rotation of the polarization of one arm of the interferometer, and there is no variation of the interferometer’s phase.

**Table 1 sensors-23-00618-t001:** Linear sensibility of Figure 4 and Figure 5.

Figure	Sensitivity	Figure	Sensitivity
Strain		Torsion	
Figure 4a	Spatial Frequency Shift (Δ*f*) (−2.1 ± 0.3) × 10^−4^ nm^−1^ µε^−1^	Figure 4c	Spatial Frequency Shift (Δ*f*) (−2.177 ± 0.002) × 10^−12^ nm^−1^/°
Figure 4b	Amplitude (Δ*A*) (4.9 ± 0.3) × 10^−7^ με^−1^	Figure 4d	Amplitude (Δ*f*) (1.02 ± 0.06) × 10^−3^/°
Figure 5a	Spectrum extreme Shift (0.576 ± 0.009) pm‧με^−1^	Figure 5b	Spectrum extreme Shift (2.09 ± 0.06) pm/°

### 3.3. For ΔL ≈ 0 and Δn ≠ 0

In a second step, the use of arms of equal length but with different refractive indices was also explored. For this, the system was initially balanced (Figure 7). An FSR of 15.5 nm at a wavelength of 1531.4 nm was obtained, which implies that there is a length difference between the interferometer arms of 0.1 mm; this is the value used as the measurement uncertainty. Two fibers with a length of 120 mm and different refractive indices were then added. An FSR of 4.5 nm at a wavelength of 1539.9 nm was obtained, which implies a Δ*n* of (4.4 ± 0.2) × 10^−3^.

## 4. Discussion

This system allows the simultaneous measurement of two physical quantities. The advantage of having two distinct arms with similar characteristics, if both arms are exposed to the same variation of a physical quantity (for example, temperature), the impact is practically null, as the optical path variations associated with each arm are identical. In addition, with a difference in arm lengths similar to the length of Hi-Bi fiber used in a traditional FLM [7], it is possible to obtain a much higher phase difference, which will allow the development of optical strain sensors that use the more sensitive Vernier effect and with a shorter fiber length, in addition to the fact that the whole system is built only with single-mode fiber. In addition, FSR control has also been demonstrated with interferometer arms of equal length but different refractive indices, which will accentuate the interferometer’s phase difference. Finally, the possibility of this new FLM configuration (ring interferometer type) allowing system balancing was demonstrated.

## 5. Conclusions

This optical fiber system allows simultaneous measurements of strain and torsion. In the case of strain, this will imply a phase variation of the interferometer, which will correspond to a variation of the spectral frequencies of (−2.1 ± 0.3) × 10^−4^ nm^−1^ με^−1^, which is 100 dB higher when compared to the torsion variation. In addition, phase variation will also result in a spectrum shift of (0.576 ± 0.009) pm‧με^−1^. As for the torsion variation, since the rotation of the polarization will occur, then there will be a cosine variation of the spectrum visibility. This can be measured by the varying amplitude of the FFT peaks, which has achieved an amplitude variation of (1.02 ± 0.06) × 10^−3^/°, which is 33 dB higher than that achieved with the strain variation. It has also been shown that the system allows the FSR to be controlled by the difference in the refraction indices of the two arms, in addition to the system’s ability to be balanced; the latter feature is not possible with traditional FLMs with Hi-bi fiber.

The ability to achieve low FSR with reduced fiber lengths will allow great strain sensitivity to be achieved with the implementation of the enhanced Vernier effect based on the push-pull method.

## Figures and Tables

**Figure 1 sensors-23-00618-f001:**
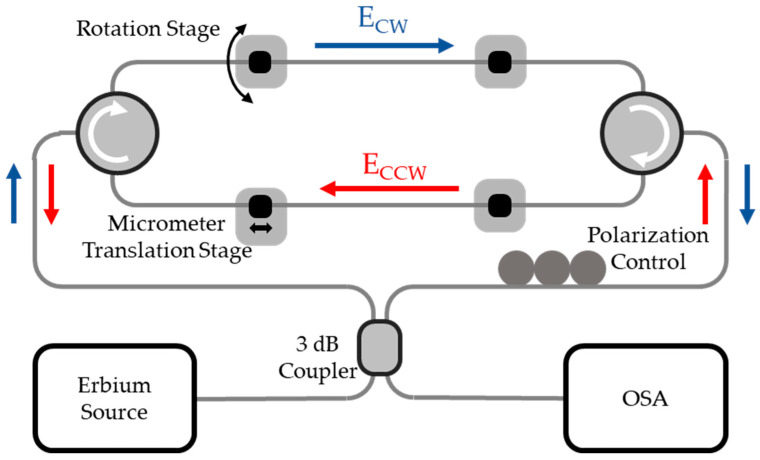
Optical system of FLM with optical circulators.

**Figure 2 sensors-23-00618-f002:**
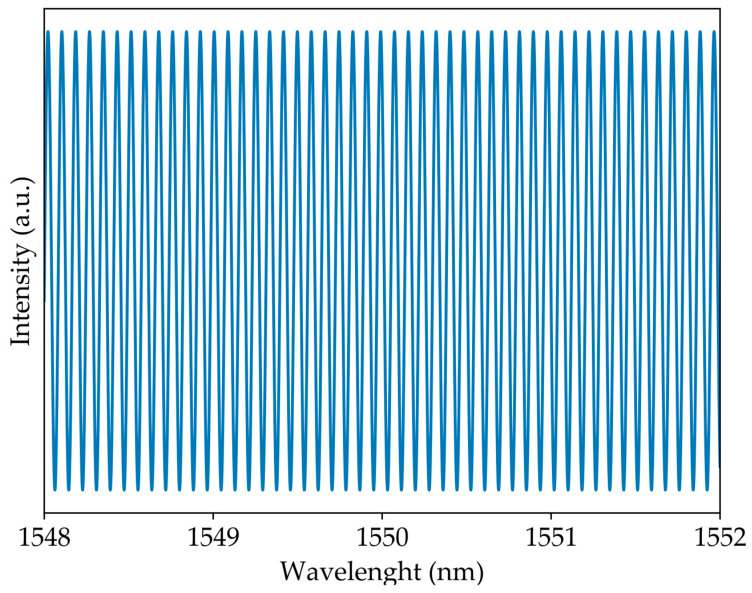
Simulation of an interferometer with Δ*L* = 20.25 mm.

**Figure 3 sensors-23-00618-f003:**
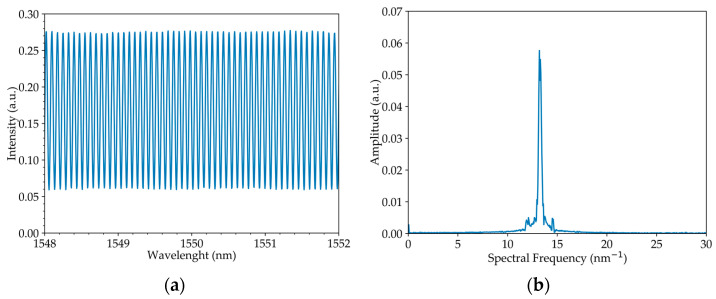
Output signal for two wave interference: (**a**) optical spectrum and (**b**) its Fourier Transform.

**Figure 4 sensors-23-00618-f004:**
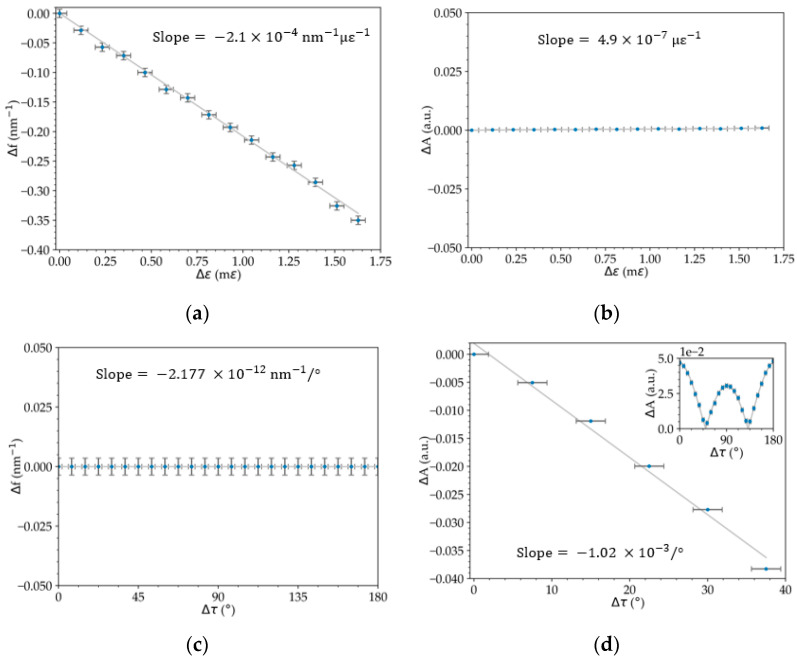
Frequency shift and its amplitude as a function of (**a**,**b**) strain and (**c**,**d**) torsion and wavelength shift of a spectrum extreme.

**Figure 5 sensors-23-00618-f005:**
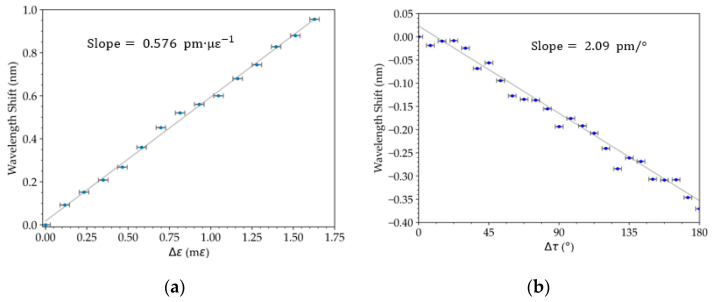
Response of the FLM in wavelength shift: (**a**) strain and (**b**) torsion.

**Figure 6 sensors-23-00618-f006:**
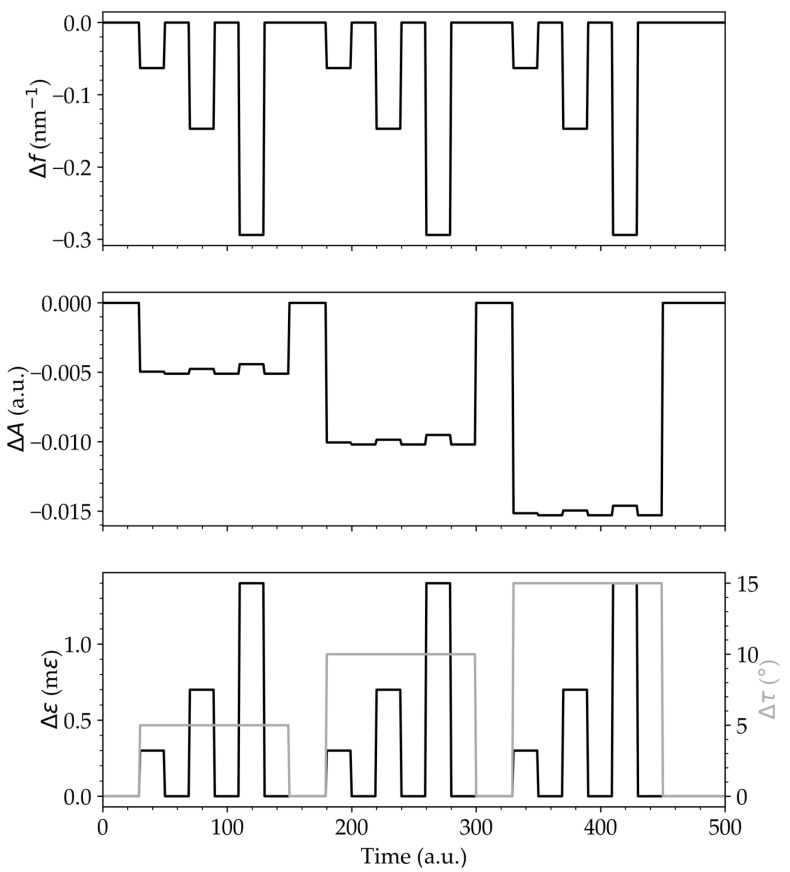
Simultaneous strain and torsion measurements by applying Equation (7).

**Figure 7 sensors-23-00618-f007:**
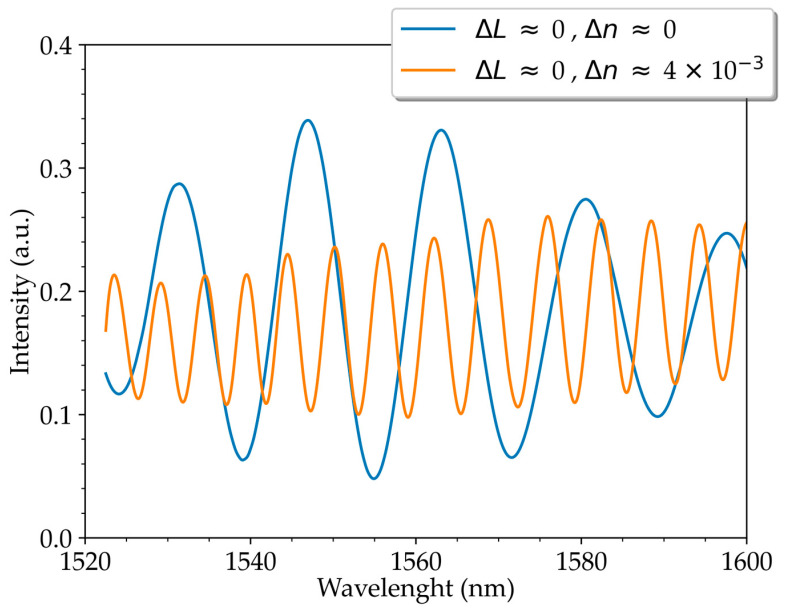
Normalized spectrum for: (**a**) balanced system (before adding two fibers of length 120 mm with different refractive indices) and (**b**) after adding two fibers of equal length, 120 mm, but different effective refractive indices.

## Data Availability

Data underlying the results presented in this paper are not publicly available at this time but may be obtained from the authors upon reasonable request.

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
