# Peer review of "Fiber Loop Mirror Based on Optical Fiber Circulator for Sensing Applications"

_sensors, 2023, doi:10.3390/s23020618_

Round 1

Reviewer 1 Report

Please check the attachment!

Author Response

Reviewer: From Line 64 to 65 and thereafter, the authors claimed: “The torsion and strain application will result the change of distinct spectrum features allowing the simultaneous measurement of the two physical quantities.”

In principle, the counterclockwise (CCW) is designed for torsion and clockwise (CW) for strain by assuming that the torsion caused polarization changes are not interfered by the path of strain and vice versa of that CW strain to the torsion-caused polarization changes.

This is a quite ideal case! We know, the strain path as a quartz must suffer from the stress birefringence, which characterizes as a polarization plane changes. So I think the result presented here is not really a simultaneous result. Even if in its presence form, it seems, can’t be employed to measure the two physical variables simultaneously, due to the designed cross reference being vulnerable.

Please provide a test data of two variables measured simultaneously!

Authors: The test data of two variables measured simultaneously was added. (lines 145 – 150)

Reviewer: Line 76 on Page 2: “... each system’s arms..” should be “...the arms of system..” ; Please check all the context carefully!

Authors: The change was implemented

Reviewer: In equation 1,? did not be explained, the symbol you used makes it like a factor rather than an operator. Maybe some space can be inserted to make it more clear.

Authors: The change was implemented. This consideration was also applied to other equation.

Reviewer 2 Report

Dear Authors, your manuscript can be accepted for publication after some minor corrections.

The quality of the figures is very low, the font size is very small, and so it the size of the numbers. The figures are too big and go out of the margins. Table 1 has no caption. My suggestions for the table is to have just two columns, one for strain and one for torsion, so both sets can be compared. The figure number can be put in the caption 

Author Response

Reviewer: The quality of the figures is very low, the font size is very small, and so it the size of the numbers. The figures are too big and go out of the margins. Table 1 has no caption.

My suggestions for the table is to have just two columns, one for strain and one for torsion, so both sets can be compared. The figure number can be put in the caption 

Authors: The quality of the images comes from the generation of the pdf from word. Due to the participation of several researchers, it was necessary to use word or instead of Latex.

The table legend was written.

The two-column table has been implemented.

Reviewer 3 Report

This paper is about the use of Fiber Loop Mirror configuration with two circulators for strain and torsion discrimination. There are some issues need to be checked and some problems to be modified.

1. There is a problem with the abbreviation of “Fiber Loop Mirror”, which should be “FLM”, not “FML”. Please modify this problem.

2. The clockwise and counterclockwise are abbreviated as CW and CCW in line 34-35, abbreviations can be used directly in line 69-70. Please check this issue.

3.There are some issues about the formulas (1)-(4):

A) Please confirm the use of italics in several formulas. Symbols are italicized in formulas but not in notes(line 79-80).

B) The formula should not be followed by a comma.

C) Some symbols in the formula are not explained. The meaning of some constants(α, β, λ) are not clear.

Please check these issues between line 76-85.

4. It is suggested to introduce the test overview in another chapter, including materials used for torsion and deformation, photos of test device, the test conditions, et al.

5. What`s the meaning of  L and  N in line 103? These two symbols are not mentioned in the above. Please explain this.

6. In Figure 4(d), Why the max  t is only 40°? But in Figure 4(c), that is 180°. And there is ambiguity between small and large graphs in Figure 4(d). Please provide an explanation.

7. Please the explanations of  f and  A before they are used in Figure 4.

8. What is the principle of Formula 5? Please provide an explanation in the Chapter of theoretical.

9. Are there any other advantages of this optical fiber system compared with other optical systems that can realize the measurement of strain and torsion? Does the system studied by the author have practical significance in practical application?Like improvement of test accuracy or automatic measurement.

10. Too many long sentences will affect readers' understanding, more short sentences are recommended. 

Author Response

Reviewer: There is a problem with the abbreviation of “Fiber Loop Mirror”, which should be “FLM”, not “FML”. Please modify this problem.

Authors: The change was implemented

Reviewer: The clockwise and counterclockwise are abbreviated as CW and CCW in line 34-35, abbreviations can be used directly in line 69-70. Please check this issue.

Authors: The change was implemented

Reviewer: There are some issues about the formulas (1)-(4):

  1. A) Please confirm the use of italics in several formulas. Symbols are italicized in formulas but not in notes (line 79-80).
  2. B) The formula should not be followed by a comma.
  3. C) Some symbols in the formula are not explained. The meaning of some constants(α, β, λ) are not clear.

Please check these issues between line 76-85.

Authors: Regarding (A) and (B), the changes have been made. For (C), italic and roman fonts were considered, α means proportionality, β is the propagation constant given by β = 2πn/λ and lambda is the wavelength. (lines 79-80).

Reviewer: It is suggested to introduce the test overview in another chapter, including materials used for torsion and deformation, photos of test device, the test conditions, et al.

Authors: The beginning of chapter 3 has been remade. It begin with a short introduction about the experimental activity and a new sub-section called test overview is added.

Reviewer: What`s the meaning of ΔL and Δn in line 103? These two symbols are not mentioned in the above. Please explain this.

Authors: In the new line 120, old line 103, ΔL is the length difference between the two arms and Δn is refractive index difference between two arms.

Reviewer: In Figure 4(d), Why the max Δτ is only 40°? But in Figure 4(c), that is 180°. And there is ambiguity between small and large graphs in Figure 4(d). Please provide an explanation.

Authors: To obtain eq. 5-7, linear variations are required. In figure 4.d we find in the inset the variation of the signal amplitude for a range of angles from 0° to 180° which follows a co-sinusoidal dependence. In the main graph only the region from 0° to 40° is focused where the variation is linear.

Reviewer: Please the explanations of Δf and ΔA before they are used in Figure 4.

Authors: In line 128, were introduced the Δf and ΔA: “In the characterization, it was determined both the frequency shift of the FFT maxima (Δf) as well as the variation of the corresponding intensities (ΔA).”

Reviewer: What is the principle of Formula 5? Please provide an explanation in the Chapter of theoretical.

Authors: An explanation was added to Theoretical Consideration. (lines 95 - 99).

Reviewer: Are there any other advantages of this optical fiber system compared with other optical systems that can realize the measurement of strain and torsion? Does the system studied by the author have practical significance in practical application? Like improvement of test accuracy or automatic measurement.

Authors: This configuration, unlike traditional FLM configurations, does not require Hi-bi fibers, being its implementation in an existing system simpler, i.e., using the information transmission fibers themselves as sensor fibers simultaneously. Furthermore, as the refractive index difference (birefringence) is achieved through two independent fibers, its value can be higher than the value of the existing Hi-bi fibers, allowing the system sensitivity to be increased: ΔΛ=Δ(nL)=n1 L1 - n2 L2

On the other hand, this system is a Sagnac interferometer, which, although it has not been demonstrated, also measures angular velocity. This system allows us to maximize the sensitivity of this measurement by calibrating the length or refractive index of the arms, in order to compensate the rotation impact and consequently the arms difference very small.

Reviewer: Too many long sentences will affect readers' understanding, more short sentences are recommended. 

Authors: The sentences were revised.

Round 2

Author Response

Reviewer: The authors presented a very superficial revision about the simultaneous measurement results as fig.6 showed as:

The results were too rural and not convinced because:

For example, to repeat the strain test given in Fig,5(a) as:

with the different torsion being applied, and contrast their difference therein.

I suggest to give them one more chance to revision!

Authors: Figure 6 has been modified based on new measurements to make the dependence of frequency and amplitude on strain and torsion clearer. As can be seen, the frequency of the spectral fringes only changes with the strain variation. The amplitude of the spectral fringes changes with both torsion and strain and is strongly dependent on the torsion in comparison to its variation with the strain. It is therefore possible, using eq. 7, to achieve simultaneous measurement of tension and torsion. The next lines were added:

“ In this system, the frequency variation of the spectral fringes occurs only with the variation of the strain. The amplitude variation depends on both the torsion and the strain, being strongly dependent on the torsion when compared to its dependence on the strain. Thus, with the application of eq. 7, it is possible to achieve simultaneous strain and torsion measurement.” (lines 150 – 154)

Reviewer 3 Report

This paper is very intersting, it is revised based on the comment. There are some questions.

(1) The linear ift can be given in the figure 4.

(2) Please give more informaiton for the figure 4(d), the small figure in figure 4(d) is important.

(3) Figure 6 has two figures, the first one should be given the X-axis informaiton.

(4) Figure 7 shows that L=0?please give the explation.

Author Response

This paper is very interesting, it is revised based on the comment. There are some questions.

  • The linear fit can be given in the figure 4.

Authors: The slopes were written in the respective figures. The fitting curves are also on the graph and refer to r2 on lines 130 - 132.

  • Please give more information for the figure 4(d), the small figure in figure 4(d) is important.

Authors: The following lines have been changed:

“where Ki,j with i = ε,τ and j = f,A are the linear fit slope, that for this system, are shown on Table 1. So, system equation is” (lines 98 – 99)

“and an amplitude variation of (1.02 ± 0.06)×10-3 /° for a range of 0º to 45° are calculated (Figure 4d). In addition, a cosine amplitude variation for a range of 0° to 180° (Figure 4d) is obtained, an expected variation with the linear polarization rotation.” (lines 140 -142)

  • Figure 6 has two figures, the first one should be given the X-axis information.

Authors: The names of the X-axes in figure 6 have been updated in accordance with the axes in figure 4.

  • Figure 7 shows that ΔL=0? please give the explanation.

Authors: As it is written at the beginning of page 7, the arms of the system have the same length, initially it was balanced. Of course, the perfectly balanced state was not achieved and therefore the spectrum still revealed bangs. Therefore, the difference between the arms, i.e., dL = 0.1 mm was determined. Next, two optical fiber sections with equal lengths but different refractive indices were joined, one to each arm of the system, and the refractive index of one of the optical fiber sections was calculated.